# Characterization of the chloroplast genome of *Lonicera ruprechtiana* Regel and comparison with other selected species of Caprifoliaceae

Lei Gu[1]☯, Yunyan Hou[1]☯, Guangyi Wang[1], Qiuping Liu[1], Wei Ding[2], Qingbei Weng ORCID[1,3]*

**1** School of Life Sciences, Guizhou Normal University, Guiyang, China, **2** Colleage of plant protection, Southwest University, Chongqing, China, **3** Qiannan Normal University for Nationalities, Duyun, China

☯ These authors contributed equally to this work.
* wengqingbei@gznu.edu.cn

**Data Availability Statement:** The annotated chloroplast genome data that support the findings of this study are openly available in GenBank of NCBI at https://www.ncbi.nlm.nih.gov under the

## Abstract

*Lonicera ruprechtiana* Regel is widely used as a greening tree in China and also displays excellent pharmacological activities. The phylogenetic relationship between *L. ruprechtiana* and other members of Caprifoliaceae remains unclear. In this study, the complete cp genome of *L. ruprechtiana* was identified using high-throughput Illumina pair-end sequencing data. The circular cp genome was 154,611 bp long and has a large single-copy region of 88,182 bp and a small single-copy region of 18,713 bp, with the two parts separated by two inverted repeat (IR) regions (23,858 bp each). A total of 131 genes were annotated, including 8 ribosomal RNAs, 39 transfer RNAs, and 84 protein-coding genes (PCGs). In addition, 49 repeat sequences and 55 simple sequence repeat loci of 18 types were also detected. Codon usage analysis demonstrated that the Leu codon is preferential for the A/U ending. Maximum-likelihood phylogenetic analysis using 22 Caprifoliaceae species revealed that *L. ruprechtiana* was closely related to *Lonicera insularis*. Comparison of IR regions revealed that the cp genome of *L. ruprechtiana* was largely conserved with that of congeneric species. Moreover, synonymous (Ks) and non-synonymous (Ka) substitution rate analysis showed that most genes were under purifying selection pressure; *ycf3*, and some genes associated with subunits of NADH dehydrogenase, subunits of the cytochrome b/f complex, and subunits of the photosystem had been subjected to strong purifying selection pressure (Ka/Ks < 0.1). This study provides useful genetic information for future study of *L. ruprechtiana* evolution.

## Introduction

*Lonicera* is one of the larger genera in Caprifoliaceae, of which approximately 200 species have been identified. Among these species, approximately 100, 30, and 25 species can be found in China, Korea, and Japan, respectively [1]. Many *Lonicera* species are rich in medicinal components and are often used as herbal medicine to treat pharyngodynia, headache, respiratory infection, and acute fever [2,3]. For example, *Lonicera japonica* Thunb. (called *Jinyinhua* in

accession number MW296954. The associated BioProject, SRA, and Bio-Sample numbers are PRJNA682877, SRX9639378, and SAMN17013271, respectively.

**Funding:** This study was supported by the Provincial Program on Platform and Talent Development of the Department of Science and Technology of Guizhou China (grant number, [2019]5655 and [2019]5617) to QW, the National Natural Science Foundation of China (grant number, 32160434) to LG, the Guizhou Provincial Science and Technology Foundation (grant number, 2020–1Y096) to LG, and the Key Project from China National Tobacco Corporation (grant number, 110202001024 (JY-07)) to QL and WD. The funders had no role in study design, data collection and analysis, decision to publish, or preparation of the manuscript.

**Competing interests:** The authors have declared that no competing interests exist.

Chinese) is an important traditional Chinese medicinal herb widely used in both the food and pharmaceutical industries [4,5]. In the Chinese Pharmacopoeia (2015), four *Lonicera* species, *L. hypoglauca*, *L. confuse*, *L. fulvotomentosa*, and *L. macranthoides*, have been recorded as Flos Lonicerae (*Shanyinhua* in Chinese). Many bioactive components, including chlorogenic acid, have been isolated and characterized from *Lonicera* species. These compounds possess a wide range of bioactive properties, such as anti-pyretic, antioxidant, and anti-hyperlipidemic properties [6]. Antioxidant activity is the fundamental driver of the pharmacological properties of *Lonicera* species [4].

*Lonicera ruprechtiana* Regel is widely distributed in the east of the three provinces of Northeast China. Because of its excellent cold and drought resistance, it is often used as a greening tree species in northern China. *L. ruprechtiana* exhibit antibacterial effects that are no weaker than those of *L. japonica* and in some aspects superior [7]. Moreover, the biological activities and therapeutic effects of *L. ruprechtiana* are similar to those of *L. japonica* making it a possible substitute for *L. japonica* [7].

The chloroplast (cp) genome is derived from the maternal parent and tends to exhibit a more highly conserved genomic structure than the nuclear genome [8]. In most plant species, the cp genome exhibits a typical quadripartite structure containing three parts, namely, inverted repeat (IR) regions (IRA and IRB), one large single-copy (LSC) region, and one small single-copy (SSC) region. The IR regions separate the LSC and SSC regions [9]. The size of the cp genome varies from 72 to 217 kb and includes about 130 genes [9]. Linear genomes have also been found in some plant species [10]. Mutational events, including insertions and deletions, inversions, substitutions, genome rearrangements, and translocations, have also been found in cp genomes [11–13]. Increasingly, studies are using polymorphism in cp genomes to explore taxonomic and phylogenetic discrepancies [14]. Cp genomes can also be used to produce vaccines through transgenic technology [15] and have become a useful and powerful tool for revealing plant phylogenies [16].

The evolution of many cp genomes in Caprifoliaceae has recently been reported [17,18]. Although *L. ruprechtiana* shows good economic and ornamental value, little genetic or genomic research has been done on this species, and the full cp genome of *L. ruprechtiana* is still not available in databases. In this work, using an Illumina sequencing platform, we assembled *de novo* the complete cp genome of *L. ruprechtiana*. We also downloaded the cp genomes of other members of Caprifoliaceae from public databases and explored the phylogenetic relationships between *L. ruprechtiana* and other related species. These data will be a useful resource for future genetic studies of *L. ruprechtiana*.

## Materials and methods

### Plant materials and sequencing

Plant leaf samples were collected from the School of Traditional Chinese Medicine, JILin Agriculture Science and Technology College, Jilin, Jilin Province, China (44˚3′5.44″N, 126˚6′34.44″E, 237 m above sea level). The leaf specimen (accession number: GL202001001) was deposited in the herbarium of the School of Life Sciences, Guizhou Normal University. Total genomic DNA (No. GL202001002) was extracted using a DNAsecure Plant Kit (TIANGEN, Beijing) and stored at -80˚C in the laboratory (room number: 1403) of the School of Life Sciences, Guizhou Normal University. A total concentration of 700 ng DNA served as the input material for the DNA sample preparations. Sequencing libraries were generated using the NEB Next® Ultra DNA Library Prep Kit for Illumina® (NEB, Ipswich, MA, USA), following the manufacturer's recommendations, and index codes were added to attribute sequences to each sample. Briefly, the DNA was purified

using AMPure XPsystem (Beckman Coulter, Beverly, USA). After the adenylation of 3' ends of DNA fragments, the NEB Next Adaptor with a hairpin loop structure was ligated to prepare for hybridization. Electrophoresis was used to select DNA fragments at a specified length. Then 3 µL USER Enzyme (NEB, USA) was used with size-selected, adaptor-ligated DNA at 37˚C for 15 min followed by 5 min at 95˚C before PCR. Then PCR was performed with Phusion High-Fidelity DNA polymerase, Universal PCR primers, and Index (X) Primer. The PCR products were purified (AMPure XP system), and the library quality was assessed on an Agilent Bioanalyzer 2100 system.

Clustering of index-coded samples was performed on a cBot Cluster Generation System using an Illumia Cluster Kit according to the manufacturer's instructions. After cluster generation, library preparations were sequenced on an Illumina platform, and 150 bp paired-end reads were generated. All of the raw data are available at NCBI (https://www.ncbi.nlm.nih.gov/). The associated BioProject, SRA, and Bio-Sample numbers are PRJNA682877, SRX9639378, and SAMN17013271, respectively.

## Genome assembly and gene annotation

After removing the sequencing adapters and low-quality reads with QC values less than 20%, Kraken2 (https://ccb.jhu.edu/software/kraken2/) was used to identify mitochondrial and cp sequences in the clean reads; then metaSPAdes [19] was used to assemble the clean reads. The assembled reads were compared to the complete cp genome of *Lonicera sachalinensis* (GenBank accession: MH028742) using BLASTn (E-value: $10^{-6}$) [20] and BLAST+ (Version, 2.9.0) to correct deviations. The assembled cp genome was annotated using GeSeq [21]. A circular gene map of the *L. ruprechtiana* cp genome was generated using OGDraw v1.2 [22]. Finally, the validated complete cp genome sequence was submitted to GenBank under the accession number MW296954.

## Repeat sequence and simple sequence repeat detection

The REPuter (https://bibiserv.cebitec.uni-bielefeld.de/reputer/) [23] was used to identify repeat sequences with the following parameters: minimal repeat size to 30, maximum computed repeats to 50, and hamming distance to 10. Match direction included forward, palindrome, reverse, and complement repeat types. To detect simple sequence repeats (SSRs) in the genome, MISA (https://webblast.ipk-gatersleben.de/misa/) [24] was used.

## Codon usage analyses

The CodonW1.4.2 program (http://downloads.fyxm.net/CodonW-76666.html) was used to calculate the synonymous codon usage of protein-coding genes (PCGs) with default settings.

## Phylogenetic analyses

Phylogenetic analyses were performed using the de novo *L. ruprechtiana* cp genome and 22 cp genomes from across the Caprifoliaceae (14 *Lonicera* species, 3 *Patrinia* species, 2 *Dipelta* species, 1 *Triosteum* species, 1 *Weigela* species, and 1 *Heptacodium* species) (S1 Table). All complete cp genomes were downloaded from NCBI (https://www.ncbi.nlm.nih.gov/). Only the homologous CDs (coding gene sequences) were used to construct phylogenetic tree to reduce data redundancy. A total of 68 homologous CDs (S2 Table) were used to determine phylogenetic relationships. Phylogenetic trees were constructed using the maximum-likelihood (ML) method (Model: Jones-Taylor-Thornton) with 1000 bootstrap replicates using MEGA7 [25].

## Whole cp genome sequence comparisons of *L. ruprechtiana*, *L. ferdinandi*, *L. vesicaria*, *L. maackii*, and *L. insularis*

To provide comprehensive information on cp sequence divergence, the *L. ruprechtiana* cp genome was compared to four other *Lonicera* genomes. The divergence of the LSC/IRB/SSC/IRA boundary regions was visualized by IRscope (https://irscope.shinyapps.io/irapp/), based on the annotations of their available cp genomes in GenBank. In addition, the mVISTA program (http://genome.lbl.gov/vista/mvista/submit.shtml) was used to compare to divergence across entire cp genomes with default settings (window size, 100bp; RepeatMasker, do not mask; RankVISTA probability threshold, 0.5).

## Synonymous and non-synonymous substitution rate calculations

To obtain the synonymous (Ks) and non-synonymous (Ka) substitution rates, we performed pairwise comparisons of the 77 protein-coding genes between the *L. ruprechtiana* cp genome and four closely related *Lonicera* species. Pairwise alignments of the common genes among species were carried out using MAFFT [26], and the Ka/Ks ratios were calculated using the KaKs_calculator 2.0 [27], with the default parameters for plant plastid code.

# Results

## Cp genome assembly and genome features

The Illumina sequencing platform produced 3,059 Mb raw data. After identifying mitochondrial and chloroplast sequences using Kraken2, 1,525,022 organellar reads were acquired. After filtering, 2,268 Mb clean reads with a Q20 value of 96.6% were obtained. The metaSPAdes was used to assemble the clean reads. According to the assembly results, there were 548 non-redundant contigs totaling 1,060,153 bp in length and with an N50 of 6,924 bp. Further analysis of the assembly results based on the reference genome (*Lonicera sachalinensis*, the reference sequences used for assembly the cp genome of *L. ruprechtiana*) using BLASTn, we got a single contig. Then, we used the BLAST+ (Version, 2.9.0) software to pairwise alignment between *L. ruprechtiana* and the corresponding reference genome (S1 Fig). As shown in S1 Fig, the genomes mostly preserve synteny, however there was one large inverted repeat (IRA and IRB regions, Fig 1); consistent with earlier research about mammalian evolution [28]. The complete cp genome sequence of *L. ruprechtiana* was 154,611 bp in size, containing an LSC region of 88,182 bp and an SSC region of 18,713 bp (Fig 1; Table 1), separated by a pair of inverted repeats (IRA and IRB) regions of 23,858 bp each (S1 Fig; Fig 1; Table 1). The total GC content of the cp genome was 38.4%; IR regions had the highest GC content, 43.4%, followed by 36.9% in the LSC region, whereas the SSC region exhibited the lowest GC content, 33% (Table 1). The genome contained 131 genes (113 unique genes), including 84 PCGs, 8 rRNA, and 39 tRNA genes (Table 1). Among the assembled genes, all rRNAs, 5 PCGs (*rps7*, *rpl2*, *ndhB*, *ycf2*, and *ycf15*), and 7 tRNAs (*trnA-UGC*, *trnI-CAU*, *trnI-GAU*, *trnL-CAA*, *trnN-GUU*, *trnR-ACG*, and *trnV-GAC*) occurred in two copies (Tables 1 and 2), and 1 tRNA (*trnG-GCC*) occurred in three copies (Tables 1 and 2). Furthermore, out of 131 genes, 84 and 13 were found in the LSC and SSC regions, respectively, while 17 genes were duplicated in the IR regions (Fig 1). In addition, 18 genes contained one intron, whereas *rps12* and *ycf3* included two introns (Table 2). Intron-exon analyses showed that the majority (110 genes, 84%) of genes had no introns, whereas 21 (16%) had them (Table 2).

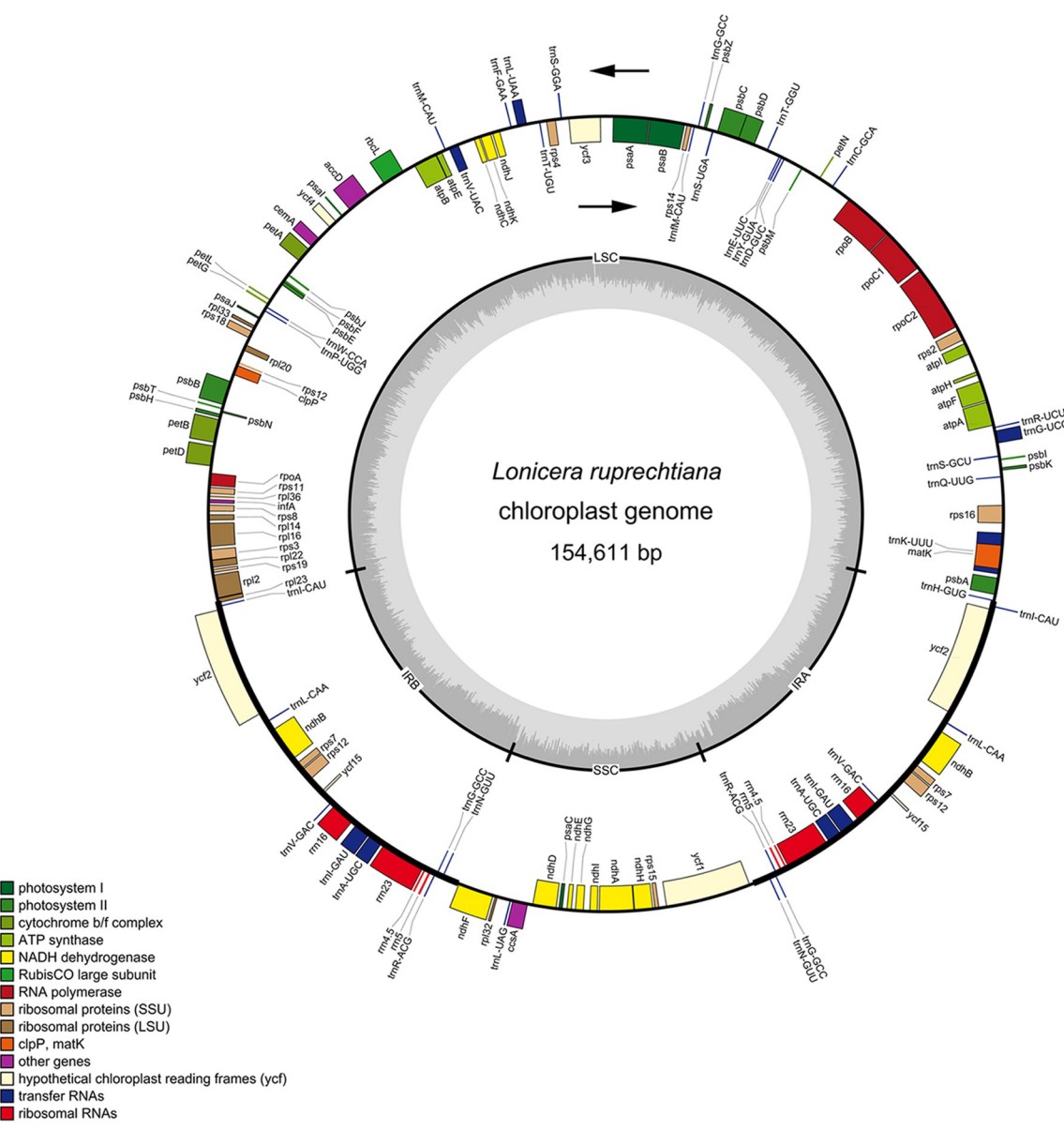

**Fig 1. Schematic diagram of the chloroplast genome of *Lonicera ruprechtiana*.** Genes on the outside and inside of the circle are transcribed in the clockwise and counter-clockwise directions, respectively. Genes belonging to different functional groups are color coded. Color intensity corresponds to GC content. The thick lines represent inverted repeat regions (IRA and IRB) that separate the LSC and SSC regions.

## SSR and repeat-sequence analyses

MISA revealed 55 SSR loci of 18 different types (of lengths of at least 10 bp) in the cp genome of *L. ruprechtiana*, which included 36 mono-, 6 di-, 2 tri-, 9 tetra-, and 2 hexanucleotide repeats (Table 3). Mononucleotide repeats were the highest percentage, containing 36 SSR motifs (65.45%) of three nucleotide types (A/T/C) (Table 3). There were six dinucleotide repeats with four different types (AT/TA/TC/GA), two trinucleotide repeats with two types (AAT/TTC), nine tetranucleotide repeats with seven types (CAAT/TTAA/ATTT/AGAT/ ATAA/TATC/TCTT), and two hexanucleotide repeats with two types (TGTTTA/ CTTACC)

Table 1. Characteristics of the complete chloroplast (cp) genome of *L. ruprechtiana*.

| Category | Items | Descriptions |
|---|---|---|
| Lengths of major regions | LSC region (bp) | 88,182 |
| | SSC region (bp) | 18,713 |
| | IRA region (bp) | 23,858 |
| | IRB region (bp) | 23,858 |
| | Total genome Size (bp) | 154,611 |
| Gene content | Total gene numbers | 131 |
| | Protein-coding gene numbers tRNA gene numbers | 84 |
| | rRNA gene numbers | 39 |
| | Two copy genes | 8 |
| | Three copy genes | 16 |
| | Genes on LSC region | 1 |
| | Genes on SSC region | 84 |
| | Genes on IRA region | 13 |
| | Genes on IRB region | 17 |
| | Gene total length (bp) | 17 |
| | Average of genes length (bp) | 77,748 |
| | Gene length / Genome (%) | 593 |
| | | 50.29% |
| GC content (%) | GC content of LSC region (%) | 36.9% |
| | GC content of SSC region (%) | 33% |
| | GC content of IRA region (%) | 43.4% |
| | GC content of IRB region (%) | 43.4% |
| | Total GC content (%) | 38.4% |

(Table 3). The longest SSR types in length were two hexanucleotide repeats (TGTTTA/ CTTACC) both of 18 bp (Table 3).

We detected 49 repeats sequences, including 18 palindromic and 31 forward repeats with lengths ranging from 55 bp to 287 bp in *L. ruprechtiana* cp genome (S3 Table). Within 49 sequences, 7, 10, 12, 5, and 7 repeats were ranging from 50–59 bp, 60–69 bp, 70–79 bp, 80–89 bp, and 90–99 bp, respectively, moreover, 8 were longer than 100 bp in length (S3 Table). In all, 30 repeats (61.22%), including 18 forward and 12 palindromic repeats, were located in *ycf2* (S3 Table). This result indicates that *ycf2* is a pseudogene. Most of repeats (about 75.51%) were contained in four protein-coding genes (*rps18*, *accD*, *ycf1*, and *ycf2*), whereas the other repeats were also found in intergenic or spacer regions (S3 Table).

## Codon usage analyses

The coding sequences of the 79 non-redundant PCGs contained 26,000 codons. Among these, leucine had the highest usage frequency, at 10.7%, while cysteine was least frequent, at only 1.1% (S4 Table; Fig 2A). To understand the synonymous codon usage bias of the *L. ruprechtiana* cp genome, the relative synonymous codon usage (RSCU) value was calculated. Thirty-one codons had RSCU values were larger than 1 (RSCU>1), suggesting that these amino acid codons was preferentially utilized by *L. ruprechtiana*. Among these, the third base of most codons were A (41.9%) or U (51.6%), with the exception of two codons, which ended with G (AUG and UUG) (S4 Table; Fig 2B).

## Phylogenetic analyses and whole cp genome sequence comparisons of *L. ruprechtiana*, *L. ferdinandi*, *L. vesicaria*, *L. maackii*, and *L. insularis*

To further understand the phylogenetic placement of *L. ruprechtiana*, 68 homologous protein-coding genes of 22 Caprifoliaceae cp genome sequences downloaded from NCBI were used to estimate a phylogeny using MEGA7 [25] with 1,000 bootstrap replicates (Fig 3). We used

**Table 2. Summary of assembled gene functions of *L. ruprechtiana* cp genome.**

| Category for genes | Groups of genes | Name of genes |
|---|---|---|
| **Self-replication** | Ribosomal RNA | *rrn4.5*[A], *rrn5*[A], *rrn16*[A], *rrn23*[A] |
| | Transfer RNA | *trnA-UGC*[A,C], *trnC-GCA*, *trnD-GUC*, *trnE-UUC*, *trnF-GAA*, *trnfM-CAU*, *trnG-GCC*[B], *trnG-UCC*[C], *trnH-GUG*, *trnI-CAU*[A], *trnI-GAU*[A,C], *trnK-UUU*[C], *trnL-CAA*[A], *trnL-UAA*[C], *trnL-UAG*, *trnM-CAU*, *trnN-GUU*[A], *trnP-UGG*, *trnQ-UUG*, *trnR-ACG*[A], *trnR-UCU*, *trnS-GCU*, *trnS-GGA*, *trnS-UGA*, *trnT-GGU*, *trnT-UGU*, *trnV-GAC*[A], *trnV-UAC*[C], *trnW-CCA*, *trnY-GUA* |
| | Small subunit of ribosome | *rps2*, *rps3*, *rps4*, *rps7*[A], *rps8*, *rps11*, *rps12*[A,C], *rps14*, *rps15*, *rps16*[C], *rps18*, *rps19* |
| | Large subunit of ribosome | *rpl2*[C], *rpl14*, *rpl16*[C], *rpl20*, *rpl22*, *rpl23*, *rpl32*, *rpl33*, *rpl36* |
| | DNA-dependent RNA polymerase | *rpoA*, *rpoB*, *rpoC1*[C], *rpoC2* |
| **photosynthesis** | Subunits of photosystem | *psaA*, *psaB*, *psaC*, *psaI*, *psaJ*, *psbA*, *psbB*, *psbC*, *psbD*, *psbE*, *psbF*, *psbH*, *psbI*, *psbJ*, *psbK*, *psbM*, *psbN*, *psbT*, *psbZ* |
| | Large subunit of Rubisco | *rbcL* |
| | Subunits of ATP synthase | *atpA*, *atpB*, *atpE*, *atpF*[C], *atpH*, *atpI* |
| | Subunits of cytochrome b/f complex | *petA*, *petB*[C], *petD*[C], *petG*, *petL*, *petN* |
| | Subunits of NADH dehydrogenase | *ndhA*[C], *ndhB*[A,C], *ndhC*, *ndhD*, *ndhE*, *ndhF*, *ndhG*, *ndhH*, *ndhI*, *ndhJ*, *ndhK* |
| **Others** | Subunit of acetyl-CoA | *accD* |
| | C-type cytochrome synthesis | *ccsA* |
| | Envelope membrane protein | *cemA* |
| | Translational initiation factor | *infA* |
| | Protease | *clpP* |
| | Maturase | *matK* |
| **unknown** | Conserved open reading frames | *ycf1*, *ycf2*[A], *ycf3*[D], *ycf4*, *ycf15*[A] |

[A, B, C, D] indicate genes with two copes, three copes, harboring one or two introns, respectively.

*Weigela* as the root, as in other phylogenetic analyses of Caprifoliaceae prior to this study [17,18]. To show the relationships between *L. ruprechtiana* and other 22 family members, the phylogenetic tree is a cladogram and branch lengths we not infered (Fig 3). The members of the Caprifolieae (*Lonicera*, *Triosteum*, and *Heptacodium*) form a clade sister to a clade containing *Dipelta* and *Patrinia*, with *Weigela* as the outgroup (Fig 3). As shown in Fig 3, *L. ruprechtiana* was sister to *L. insularis* with a bootstrap support of 100%.

To investigate divergence in the cp genome between *L. ruprechtiana* and the four other closely related species (*L. ferdinandi*, *L. vesicaria*, *L. maackii*, and *L. insularis*) (Fig 3), multiple alignments of the five cp genomes were performed. Sequence identities were plotted using mVISTA with reference to the annotation of *L. ruprechtiana* (Fig 4). All five cp genomes displayed a high sequence similarity (>85%), but several short and long inserted regions were also observed (Fig 4). Our results show that coding regions were more conserved than non-coding regions (Fig 4), consistent with earlier research [29]. The most highly conserved cp genes were the rRNA and tRNA genes (Fig 4), and the most divergent genes were *ycf1*, *accD*, *ycf2*, *rpl14*, and *atpA* (Fig 4).

The IR regions were responsible for the size variation in the cp genome. Comprehensive comparison at the LSC/IR/SSC boundaries was performed among five *Lonicera* species (Fig 5). The LSC/IRB and IRA/LSC border regions were relatively more conservative than SSC/IRA and IRB/SSC junctions (Fig 5). The *rp123* gene was present in the LSC/IRB junction; 170 bp was present in the LSC part, and 121 bp was present in the IRB part of all five cp genomes (Fig 5). The *ndhF* and *ycf1* genes were located in the SCC parts in all cp genomes, with the only

Table 3. Summary of simple sequence repeats (SSRs) in *L. ruprechtiana* cp genome.

| Repeat Unit | Motif Type | Number | Longest Repeat (bp) |
|---|---|---|---|
| 1 | A | 18 | 15 |
| | T | 17 | 14 |
| | C | 1 | 10 |
| 2 | AT | 2 | 10 |
| | TA | 2 | 12 |
| | TC | 1 | 10 |
| | GA | 1 | 10 |
| 3 | AAT | 1 | 12 |
| | TTC | 1 | 12 |
| 4 | CAAT | 1 | 16 |
| | TTAA | 1 | 16 |
| | ATTT | 1 | 12 |
| | AGAT | 2 | 12 |
| | ATAA | 2 | 12 |
| | TATC | 1 | 12 |
| | TCTT | 1 | 12 |
| 6 | TGTTTA | 1 | 18 |
| | CTTACC | 1 | 18 |
| Total | 18 | 55 | — |

difference being the distance of two genes to the junction (Fig 5). For IRB/SSC boundaries, distances of 92 bp, 70 bp, 44 bp, 88 bp, and 35 bp from the *ndhF* gene to the boundary were found in *L. ferdinandi*, *L. vesicaria*, *L. maackii*, *L. insularis*, and *L. ruprechtiana*, respectively (Fig 5). However, for SSC/IRA boundaries, the distances were 137 bp, 239 bp, 261 bp, 231 bp, and 232 bp from the *ycf1* gene to the boundary in *L. ferdinandi*, *L. vesicaria*, *L. maackii*, *L. insularis*, and *L. ruprechtiana*, respectively (Fig 5).

## Ks and Ka substitution rate analyses between *L. ruprechtiana* and the four closely related species (*L. ferdinandi, L. vesicaria, L. maackii*, and *L. insularis*)

Ka and Ks are important markers for evaluating selection pressure on genes and genomes [30]. The Ka/Ks values of 77 protein-coding genes were calculated between *L. ruprechtiana* and four other species (*L. ferdinandi*, *L. vesicaria*, *L. maackii*, and *L. insularis*) (S5 Table). Except for no polymorphisms genes, most tested genes had a Ka/Ks value below 1, indicating extensive purifying selection (S5 Table). The *PsbJ* and *rpl32* genes had a Ka/Ks value far more than 1 (>45) in all tested comparisons, indicating that these may be pseudogenes [31,32] or may have undergone strong positive selection (S5 Table). Some genes (*atpE*, *atpF*, *matK*, *ndhB*, *petB*, *petD*, *psaI*, and *rpl14*) had different Ka/Ks values (>1 or <1) under different comparisons (S5 Table), possibly due to these species/genes have been subject to different selection regimes. Some genes (*ndhG*, *petA*, *petL*, *petN*, *psaC*, *psbA*, *psbD*, *psbE*, *psbM*, *rpl36*, and *rps16*) had Ka/Ks values below 0.1 in different comparisons (for *ycf3* it was 0.001 in all tested comparisons), indicating that these genes are under strongly purifying selection (S5 Table).

## Discussion

*L. ruprechtiana* is an ornamental tree and also a potential medicinal plant. We assembled *de novo* the *L. ruprechtiana* cp genome using an Illumina sequencing platform and compared the

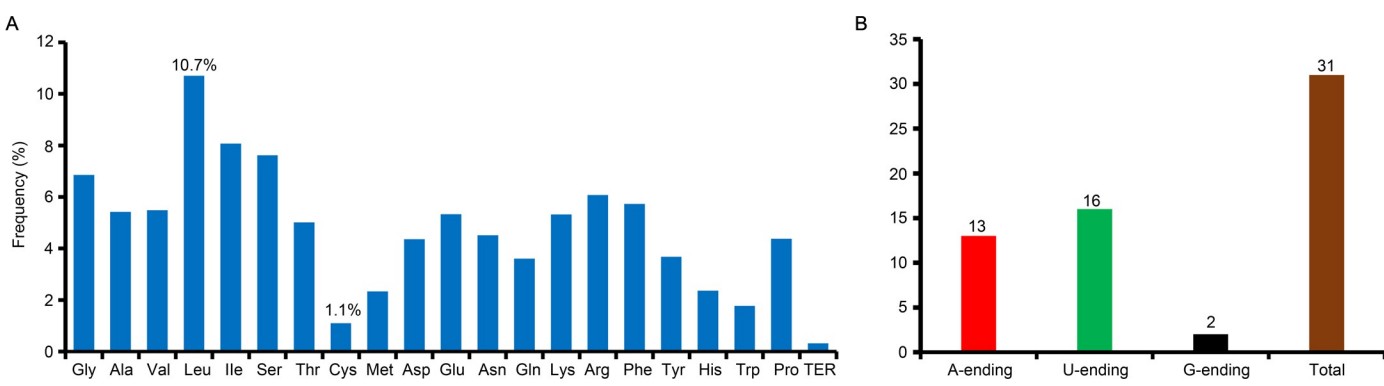

**Fig 2.** Percentage of amino acids of the *L. ruprechtiana* chloroplast (cp) genome (A) and the ending patterns of biased-usage codons (RSUC>1) (B).

cp genome with four closely related *Lonicera* species. The complete cp genome of *L. ruprechtiana* showed a typical quadripartite cycle of 154,611 bp in length (Fig 1; Table 1). The cp genomes in angiosperms have conserved features with almost the same gene content and organization [33,34]. The complete cp genome of *L. ruprechtiana* showed a typical quadripartite cycle of 154,611 bp in length, comparable to that of published *Lonicera* species cp genomes (154,513–155,346) (Fig 1; Tables 1 and S1) [17]. The variation of IR regions and boundaries in SSC/IR and LSC/IR have been thought to be critical in determining the length variation in the angiosperm cp genome [35]. This variation allowed us to explore the evolution of the cp genome [33,36]. Except for some minor variation in the distance from adjacent genes to the

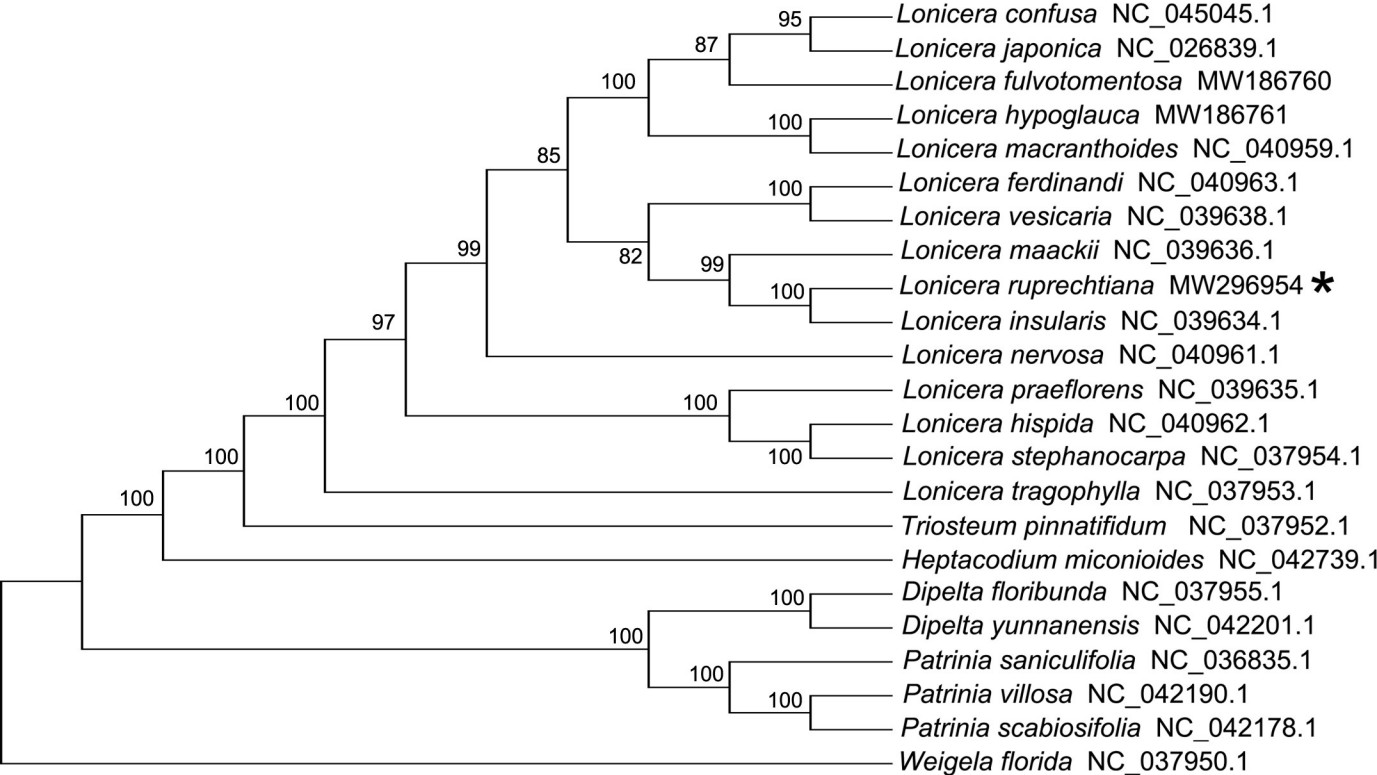

**Fig 3. Cladogram summarizing the evolutionary relationships of 23 Caprifoliaceae species based on 68 homologous protein-coding genes of the chloroplast genomes.** GenBank accession numbers are given. Shown next to the nodes are bootstrap support values based on 1,000 replicates.

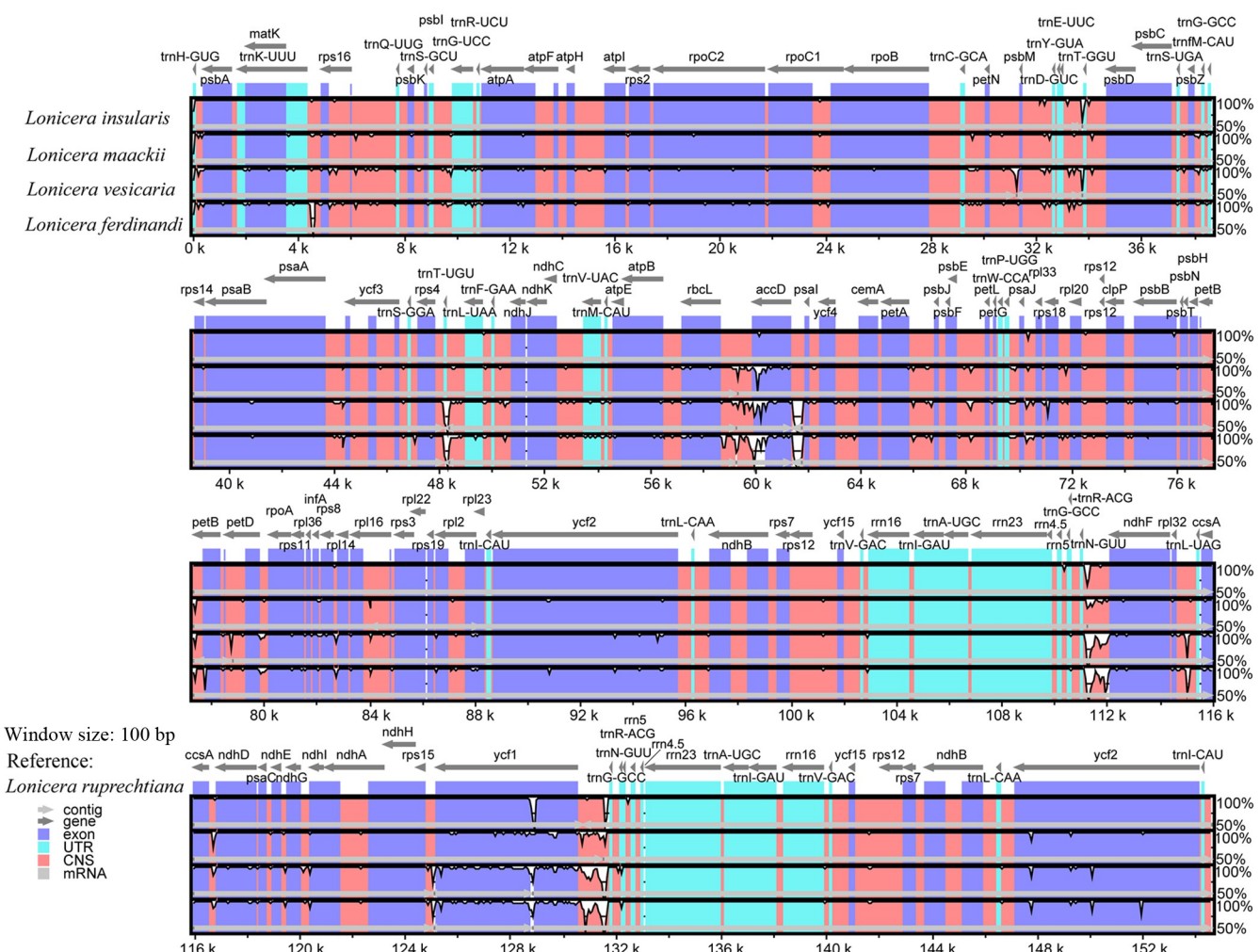

**Fig 4. Alignment of the chloroplast genome of *L. ruprechtiana* and those of four closely related species. The alignment was performed using mVISTA with *L. ruprechtiana* as a reference. Local collinear blocks within each alignment are indicated by color and linkages.**

boundaries, the sizes of the junctions of LSC, SSC, and IR of *L. ruprechtiana* were similar to those of the four closely related *Lonicera* species (Fig 5). We obtained 84 protein-coding genes in the *L. ruprechtiana* cp genome. Among the five species compared in this paper, three genes were missing in some species, including *accD* (missing in *L. ferdinandi* and *L. vesicaria*), *psbL* (missing in *L. ruprechtiana*), and *ycf15* (missing in *L. ferdinandi*).

We have obtained a total of 55 SSR motifs including 18 different types in the *L. ruprechtiana* cp genome (Table 3). The A/T type mononucleotide repeats accounted for the majority of these SSRs, similar to other reports [37–40]. The amounts of polyadenine (polyA) and polythymine (polyT) in the cp genome may be the cause for the abundance of A/T type SSRs. The SSR loci identified in this research can help us understand the population genetic structure of *L. ruprechtiana* species. Repeat sequences are thought to play an important role in the rearrangements of genome, and the variation between lineages can be used as a genomic marker for phylogenetic analysis [41,42]. A total of 49 repeat sequences of 55–287 bp were found in the *L. ruprechtiana* cp genome (S3 Table). The *ycf2* gene accounted for about half of these repeats (30 of 55) (S3 Table). Similar results have been reported for other cp genomes [43,44], indicating that *ycf2* is one of the most variable genes in the chloroplast genome.

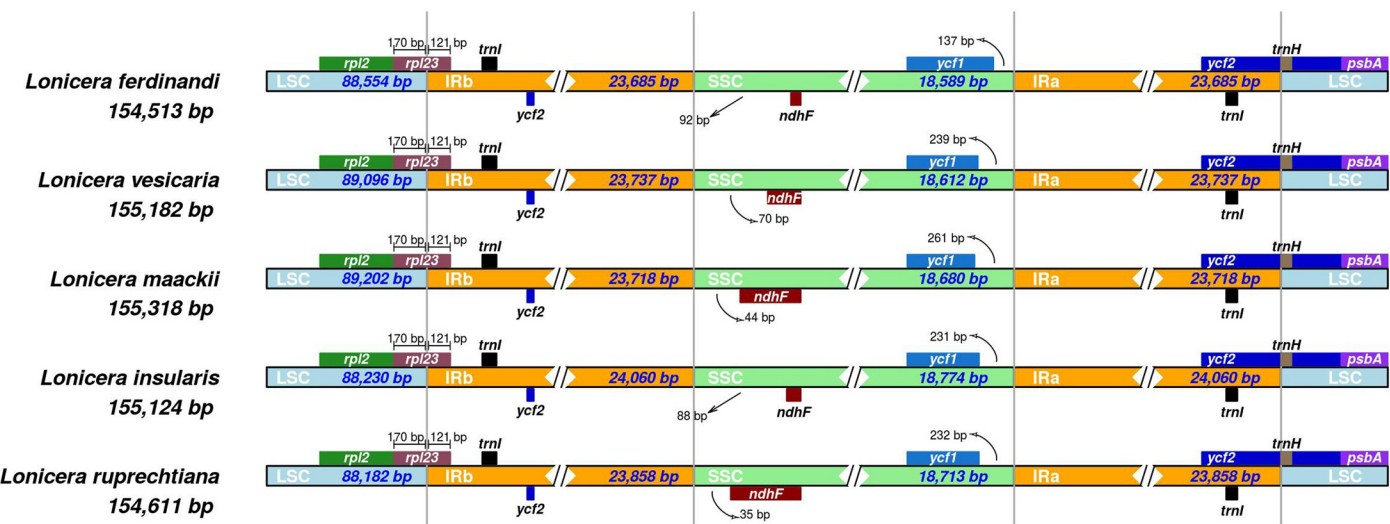

**Fig 5. Comparative analyses of the boundary regions (LSC, SSC, and IR) and adjacent genes among five chloroplast (cp) genomes.** Sequences of the LSC/IRB/SSC/IRA boundaries and adjacent genes in *L. ruprechtiana* and four closely related cp genomes (*L. ferdinandi, L. vesicaria, L. maackii,* **and** *L. insularis*) **are shown.** Genes transcribed by positive-strand are shown above the line, whereas genes that transcribed by reverse-strand are shown below the line. Gene names are indicated in boxes, and their lengths in junction sites are displayed above the boxes. Lengths (bp) represent the distances between genes and junction sites.

Codon usage is an important parameter to understand evolutionary relationships and the selection pressure acting on genes; the relatively high similarities of codon usage among different species indicated that these species may experience similar environmental stresses [45]. Among the high-usage codons (RSCU > 1) in the *L. ruprechtiana* cp genome, the preference was for codon endings with A/U (S4 Table; Fig 2B). This might be due to the number of A/T nucleotides observed in angiosperm cp genomes [33,46].

A phylogenetics tree was reconstructed based on homologous protein-coding genes for the Caprifoliaceae family, and it revealed a close relationship between *L. ruprechtiana* and *L. insularis* (Fig 3). *L. insularis* is an endemic plant found on Dokdo islet, Korea. *L. insularis* has been phytochemically investigated; an iridoid glycoside, an argininosecologanin, and six lignans have been reported from the stem of this species [47,48]. Like *L. insularis*, *L. ruprechtiana* may also contain iridoids and secoiridoids. The economic and medicinal value of *L. ruprechtiana* require further investigation.

## Conclusion

The complete *L. ruprechtiana* cp genome was assembled *de novo* using an Illumina platform. It has a typical quadripartite cycle 154,611 bp long, which includes 84 protein-coding genes, 39 tRNA genes, and 8 rRNA genes. A total of 49 repeat sequences and 55 SSR loci were identified and could be useful for phylogenetic studies and marker development. Codon usage analyses revealed that the Leu codon ending with A/U was preferentially utilized. A phylogenetic tree based on homologous protein-coding genes of 23 Caprifoliaceae family members revealed that *L. ruprechtiana* is closely related to *L. insularis*. Our findings can be used for further cp studies in *L. ruprechtiana*. In addition, our results broaden knowledge of the genome organization and evolution of Caprifoliaceae species.

## Supporting information

**S1 Fig. Pairwise alignment plots.** Red is the result in the same direction; blue is the result in the opposite direction.
(TIF)

**S1 Table. List of the 23 cp genomes used for phylogenetic analysis.**
(XLSX)

**S2 Table. The homologous CDs (coding gene sequences) used to construct phylogenetic tree.**
(XLSX)

**S3 Table. Repeat sequences of** *L. ruprechtiana* **cp genome.**
(DOCX)

**S4 Table. Codon usage analysis of protein coding genes of** *L. ruprechtiana* **cp genome.**
(XLSX)

**S5 Table. Synonymous (Ks) and non-synonymous (Ka) substitution rates of common coding genes between** *L. ruprechtiana* **and other four closed related species.**
(XLSX)

## Author Contributions

**Data curation:** Yunyan Hou.

**Formal analysis:** Lei Gu, Yunyan Hou.

**Funding acquisition:** Lei Gu, Qiuping Liu, Wei Ding.

**Investigation:** Qingbei Weng.

**Methodology:** Guangyi Wang.

**Project administration:** Qingbei Weng.

**Resources:** Yunyan Hou, Guangyi Wang.

**Software:** Lei Gu, Yunyan Hou, Qiuping Liu.

**Writing – original draft:** Lei Gu, Yunyan Hou.

**Writing – review & editing:** Yunyan Hou, Qiuping Liu, Qingbei Weng.

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
