## [Decision Letter · Decision Letter 0]

16 Sep 2021

PONE-D-21-23416Characterization of the chloroplast genome of Lonicera ruprechtiana Regel and comparison with other selected species of CaprifoliaceaePLOS ONE

Dear Dr. Weng,

Thank you for submitting your manuscript to PLOS ONE. After careful consideration, we feel that it has merit but does not fully meet PLOS ONE’s publication criteria as it currently stands. Therefore, we invite you to submit a revised version of the manuscript that addresses the points raised during the review process.

From the Reviewers' reports, it is obviuos that numerous methodological solutions were not properly chosen or at least were not satisfactorily introduced or explained. Please provide a detailed Response to Reviewers stating each of their concerns along with the intervention made in the revised version of the manuscript. Also, be aware of additional remarks provided in the attachment by Reviewer #1.==============================

We look forward to receiving your revised manuscript.

Kind regards,

Branislav T. Šiler, Ph.D.

Academic Editor

PLOS ONE

“This study was supported by Provincial Program on Platform and Talent Development of the Department of Science and Technology of Guizhou China under Grant [No. [2019]5655 and [2019]5617], and the Guizhou Provincial Science and Technology Foundation under Grant (No. 2020–1Y096).”           

“This study was supported by Provincial Program on Platform and Talent Development of the Department of Science and Technology of Guizhou China under Grant [No. [2019]5655 and [2019]5617], and the Guizhou Provincial Science and Technology Foundation under Grant (No. 2020–1Y096).”

Please note that funding information should not appear in other areas of your manuscript. We will only publish funding information present in the Funding Statement section of the online submission form.

“This study was supported by Provincial Program on Platform and Talent Development of the Department of Science and Technology of Guizhou China under Grant [No. [2019]5655 and [2019]5617], and the Guizhou Provincial Science and Technology Foundation under Grant (No. 2020–1Y096).”

5. We noticed you have some minor occurrence of overlapping text with the following previous publication, which needs to be addressed:

- https://journals.plos.org/plosone/article?id=10.1371/journal.pone.0239823

In your revision ensure you cite all your sources (including your own works), and quote or rephrase any duplicated text outside the methods section. Further consideration is dependent on these concerns being addressed

Reviewers' comments:

Reviewer's Responses to Questions

**Comments to the Author**

1. Is the manuscript technically sound, and do the data support the conclusions?

Reviewer #1: Yes

Reviewer #2: Partly

2. Has the statistical analysis been performed appropriately and rigorously? 

Reviewer #1: Yes

Reviewer #2: No

3. Have the authors made all data underlying the findings in their manuscript fully available?

Reviewer #1: Yes

Reviewer #2: Yes

4. Is the manuscript presented in an intelligible fashion and written in standard English?

Reviewer #1: Yes

Reviewer #2: Yes

5. Review Comments to the Author

Reviewer #1: This paper presents the chloroplast genome of Lonicera ruprechtiana and conducts a few comparative genomic analyses between L. rupgrechtiana and other members of Lonicera or Caprifoliaceae. It is short and well done, but it lacks novelty. I have only minor comments about the strutucture of some of the writing and suggested inline edits, which I provide in notes attached.

Reviewer #2: The manuscript reports a study that uses methods to thoroughly examine the newly sequenced plastome of Lonicera ruprechtiana in the context of other complete plastomes from Caprifoliaceae family. This is a generally valuable study, like many other complete plastome studies that are currently published.

My main concern about the paper is on the assembly stats that are not adequately provided in the results, what was the N50 and the steps involved. Did The authors obtained a single contig with metaspades? I also hadn’t seen the use of the kraker for filtering cp and mt reads.

How many reads were used to assembly the cp genome after kraker filtering. The authors really should clarify all of this issue.

I also think that PCR should have been done for verifying the junctions followed by Sanger sequencing. This would have helped in justifying the assembly.

But, if PCR was not done, I recommend the authors the needs to include a coverage analysis of the assembly to ensure accurate assembly of the chloroplast genome. Please include this important analysis in the paper.

In addition, since non-coding region are also powerful for phylogenetic resolution, were these regions tested in this case? If yes, was the result consistent with the one presented in the publication?

The authors should also clarify the number of genes used to construct the phylogenetic tree.

The figures are also low quality and really should be improved.

6. PLOS authors have the option to publish the peer review history of their article (what does this mean?). If published, this will include your full peer review and any attached files.

Reviewer #1: No

Reviewer #2: No

---

## [Author Response · Author response to Decision Letter 0]

24 Nov 2021

December 24, 2021

Dr Branislav T. Šiler

Academic Editor 

PLOS ONE 

Dear Editor and Reviewers,

RE: Response to Decision Letter for Manuscript ID. PONE-D-21-23416

First, thank you for your comments. We have now revised our manuscript further in accordance with your recommendations. 

The following is the response to the comments of the academic editor and the reviewers with a detailed and clear point-by-point response.

We are looking forward to your positive decision. 

Sincerely yours,

Qingbei Weng, 

Ph.D, Professor

Department of Biotechnology

School of Life Sciences, 

Guizhou Normal University, Guiyang, China

Manuscript ID. PONE-D-21-23416

Dear Dr. Weng,

Thank you for submitting your manuscript to PLOS ONE. After careful consideration, we feel that it has merit but does not fully meet PLOS ONE’s publication criteria as it currently stands. Therefore, we invite you to submit a revised version of the manuscript that addresses the points raised during the review process.

From the Reviewers' reports, it is obviuos that numerous methodological solutions were not properly chosen or at least were not satisfactorily introduced or explained. Please provide a detailed Response to Reviewers stating each of their concerns along with the intervention made in the revised version of the manuscript. Also, be aware of additional remarks provided in the attachment by Reviewer #1.

We look forward to receiving your revised manuscript.

Kind regards,

Branislav T. Šiler, Ph.D.

Academic Editor

PLOS ONE

Response: Thank you for your comment. We have changed our manuscript to meet PLOS ONE's style requirements.

“This study was supported by Provincial Program on Platform and Talent Development of the Department of Science and Technology of Guizhou China under Grant [No. [2019]5655 and [2019]5617], and the Guizhou Provincial Science and Technology Foundation under Grant (No. 2020–1Y096).” 

Response: Thank you for your comment. We have added the statement the role of funders took in the study and author contributions in cover letter.

“This study was supported by Provincial Program on Platform and Talent Development of the Department of Science and Technology of Guizhou China under Grant [No. [2019]5655 and [2019]5617], and the Guizhou Provincial Science and Technology Foundation under Grant (No. 2020–1Y096).”

Please note that funding information should not appear in other areas of your manuscript. We will only publish funding information present in the Funding Statement section of the online submission form.

“This study was supported by Provincial Program on Platform and Talent Development of the Department of Science and Technology of Guizhou China under Grant [No. [2019]5655 and [2019]5617], and the Guizhou Provincial Science and Technology Foundation under Grant (No. 2020–1Y096).”

Response: Thank you for your comment. We have removed any funding-related text from the manuscript and added the funding statements in cover letter.

Response: Thank you for your comment. We have included captions for Supporting Information files at the end of our manuscript, and updated any in-text citations to match accordingly.

5. We noticed you have some minor occurrence of overlapping text with the following previous publication, which needs to be addressed:

- https://journals.plos.org/plosone/article?id=10.1371/journal.pone.0239823

In your revision ensure you cite all your sources (including your own works), and quote or rephrase any duplicated text outside the methods section. Further consideration is dependent on these concerns being addressed

Response: Thank you for your comment. We have rephrased any duplicated text outside the methods section.

Reviewer(s)' Comments to Author:

Reviewer: 1

General comments

----------------

1. Phylogenetic analyses. In your phylogenetic analyses you should state more information about how the genes were chosen, how multiple sequence alignment was done, and the model of sequence evolution was chosen and applied. Some of this information is in the results (and should be moved to the methods). Lines 194-199 should be in the methods because they state which species were used and redundantly state how the tree was made. In the results you should state how many genes you used and detail information on the sequence matrices (informative sites, invariant sites, matrix size, etc.). When discussing relationships, you should state that taxa are "sister" to one another when they are each other’s closest relatives, not that they cluster together or are related. All species are related. I also point you towards a newer chloroplast phylogeney of the Dipsacales that includes multiple members of Lonicera:

Lee, A. K. et al. Reconstructing Dipsacales phylogeny using Angiosperms353: issues and insights. Am J Bot 108, 1122–1142 (2021).

Their paper assembles more Caprifoliacea cp genomes. In their analyses Weigela is the most distantly related of the taxa you survey for your phylogeny and should be used for rooting purposes. You should include information on branch length in your phylogeny, or state that it is a cladogram only to show the relationships and branch lengths we not infered. 

Response: Thank you for your important comment. We have changed phylogenetic analyses and relative result parts in the revise manuscript. The detail changes as follow:

Phylogenetic analyses: 

Phylogenetic analyses were performed between L. ruprechtiana and the other 22 cp genome sequences of the Caprifoliaceae (14 Lonicera species, 3 Patrinia species, 2 Dipelta species, 1 Triosteum species, 1 Weigela species, and 1 Heptacodium species) (S1 Table). All complete cp genomes were downloaded from NCBI (https://www.ncbi.nlm.nih.gov/). Only the homologous CDs (coding gene sequences) were used to construct phylogenetic tree to reduce data redundancy. A total of 68 homologous CDs, including psbB, psbA, ndhB, rps7, rps12, psbA, matK, rps16, psbK, psbI, atpA, atpF, atpH, atpI, rps2, rpoC2, rpoC1, rpoB, petN, psbM, psbD, psbC, psbZ, rps14, ycf3, rps4, ndhJ, ndhC, atpE, atpB, rbcL, psaL, ycf4, cemA, petA, psbJ, psbF, psbE, petL, petG, psaJ, rpl33, rpl20, psbB, psbT, psbN, psbH, petB, petD, rps11, rpl36, infA, rps8, rpl14, rpl16, rpl22, rps19, rpl2, ndhF, rpl32, ccsA, ndhD, psaC, ndhE, ndhG, ndhI, ndhA, and ndhH, were used to determine the phylogenetic relationship. Phylogenetic trees were constructed using the maximum-likelihood (ML) method (Model: Jones-Taylor-Thornton) with 1000 bootstrap replicates using MEGA7 [24].

Result part:

Phylogenetic analyses and whole cp genome sequence comparisons

To further understand the phylogenetic relationships of L. ruprechtiana, 68 homologous protein-coding genes of 22 cp genome sequences of the Caprifoliaceae were downloaded from NCBI database to build the phylogenetic tree. The phylogenetic tree was created using MEGA7 [24] with 1,000 bootstrap replicates. To show the relationships between L. ruprechtiana and other 22 family members, the phylogenetic tree is a cladogram and branch lengths we not infered (Fig. 3). As shown in Fig. 3, the cladogram clearly classified these 23 species into 2 distinct clusters. In the Lonicera cluster, L. ruprechtiana was fully resolved in a clade with L. insularis with a bootstrap support of 100% (Fig. 3), sister to three other species of L. ferdinandi, L. vesicaria and L. maackii (Fig. 3).

We also changed the figure legend of figure 3:

Figure 3. Cladogram summarizing the evolutionary relationships of 23 Caprifoliaceae species based on 68 homologous protein-coding genes of the chloroplast genomes. The maximum-likelihood tree shows the two distinct clusters. GenBank accession numbers are given in the figure. Shown next to the nodes are bootstrap support values based on 1,000 replicates.

We also cited this literature “Lee, A. K. et al. Reconstructing Dipsacales phylogeny using Angiosperms353: issues and insights” in the manuscript. The detail changes as follow:

The phylogenomic evolution of many cp genomes in Caprifoliaceae has recently been reported [17, 18]. 18: Lee, A. K. et al. Reconstructing Dipsacales phylogeny using Angiosperms353: issues and insights. Am J Bot 108, 1122–1142 (2021).

2. Comparison of L. japonica and L. ruprechtiana (L52-59). I do not know about the phytochemistry of these species but there are a few odd things in this paragraph. First I do not know what it means for L. ruprechtiana to be "a homologue to L. japonica" (L54). Homology has a clear definition in evolutionary biology and this sentence should be removed. There are multiple claims about the phytochemistry of L. ruprechtiana in comparison to L. japonica in this paragraph but none of them are cited. What is the evidence for these claims? Finally, the last sentence about introgression is not tied to the rest of the paragraph. It is not clear that a cp phylogeny alone will tell you about introgression, nor have you established a hypothesis of introgression between these species.

Response: Thank you for your important comment. We have rewritten this paragraph in the revise manuscript. We removed " L. ruprechtiana to be a homologue to L. japonica “and the last sentence of this paragraph. We also cited the phytochemistry of L. ruprechtiana in comparison to L. japonica. The detail changes as follow:

Lonicera ruprechtiana Regel is widely distributed in the east of the three provinces of Northeast China. Because of its excellent cold and drought resistance, it is often used as a greening tree species in northern China. L. ruprechtiana exhibit antibacterial effects that are no weaker than those of L. japonica and in some aspects superior [7]. Moreover, the biological activities and therapeutic effects of L. ruprechtiana are similar to those of L. japonica making it a possible substitute for L. japonica [7].

7: Zhu D, Zhao X, Dai L, Ji S. Experimental study on the pharmacology of Lonicera ruprechtiana Regel. Theory and Practice of Chinese Medicine. 2003; 2003: 111-112. (in chinese)

3. You should state the species you are comparing in the sections "Whole cp genome sequence comparisons" and "Synonymous and nonsynonymous substitution rate calculations".

Response: Thank you for your important comment. We have changed in the revise manuscript. The detail changes as follow:

Whole cp genome sequence comparisons of L. ruprechtiana, L. ferdinandi, L. vesicaria, L. maackii, and L. insularis

Synonymous and nonsynonymous substitution rate calculations of L. ruprechtiana, L. ferdinandi, L. vesicaria, L. maackii, and L. insularis

Phylogenetic analyses and whole cp genome sequence comparisons of L. ruprechtiana, L. ferdinandi, L. vesicaria, L. maackii, and L. insularis

Ks and Ka substitution rate analyses between L. ruprechtiana and the four other closely related species (L. ferdinandi, L. vesicaria, L. maackii, and L. insularis)

4. ycf2 as a pseudogene. I am not familiar with the criteria that determine a pseudogene, but in L180 you state that ycf2 is an "essential pseudogene" because it exists in many repeats in the cp genome. What makes this locus an "essential pseudogene" (I don't know what essential means in this context either) and not simply a repetitive element belonging to a larger class such as LTR, SINE, etc?

Response: Thank you for your important comment. We have removed “essential pseudogene” in the revise manuscript.

5. L208-209 you state that your research is consistent with other research, but it is not clear in what way. Do you mean that coding regions are more conserved than non-coding regions? That is our expectation of sequence evolution.

Response: Thank you for your important comment. We mean that coding regions are more conserved than non-coding regions. We have changed in the revise manuscript. The detail changes as follow:

Our results mean that coding regions are more conserved than non-coding regions (Fig. 4), are consistent with those of earlier research [28].

----------

Line edits

----------

Abstract

- L26-27: Remove "and mVISTA", and change "conserved" to "largely conserved".

Response: Thank you for your comment. We have changed in the revise manuscript.

- L29: "genes under purifying" to "gene are under purifying"

Response: Thank you for your comment. We have changed in the revise manuscript.

Introduction

- L38: "large" to "larger"

Response: Thank you for your comment. We have changed in the revise manuscript.

- L43: Remove "(L. japonica)"

Response: Thank you for your comment. We have changed in the revise manuscript.

- L45-46: Remove "including" and place species in parentheses with "L." instead of "Lonicera"

Response: Thank you for your comment. We have changed in the revise manuscript.

- L60-61: Restructure. "The chloroplast (cp) genome is derived from the maternal parent and tends to exhibit a more highly conserved genomic structure than the nuclear genome."

Response: Thank you for your comment. We have changed in the revise manuscript.

Materials and methods

- L135: "was compared to four cp genomes in the Lonicera genus." to "was compared to four other Lonicera cp genomes."

Response: Thank you for your comment. We have changed in the revise manuscript. 

- L139: Remove "among these related species."

Response: Thank you for your comment. We have changed in the revise manuscript.

Results

- L186: "Among these, the leucine codons had" to "Among these, leucine had"

Response: Thank you for your comment. We have changed in the revise manuscript.

- L187: "of cysteine codons was" to "of cystine was"

Response: Thank you for your comment. We have changed in the revise manuscript.

- L207-208: Remove ", indicating that the L. ruprechtiana cp genome underwent evolutionary divergence."

Response: Thank you for your comment. We have changed in the revise manuscript.

- L210: "conserved cp genes are" to "conserved cp genes were"

Response: Thank you for your comment. We have changed in the revise manuscript.

- L211: "obviously" to "most"

Response: Thank you for your comment. We have changed in the revise manuscript.

- L215: "relatively conserved" to "relatively more highly"

Response: Thank you for your comment. We have changed in the revise manuscript.

- L218: "being in the" to "being the"

Response: Thank you for your comment. We have changed in the revise manuscript.

- L226-228: Ka/Ks is only calculated relative to another sequence, so you should rephrase this sentence to state that Ka/Ks were calculated between L. ruprechtiana and the four other species. What are "sequence consistent genes"? Are these genes that had no polymorphisms? This sould be restated.

Response: Thank you for your comment. We have changed in the revise manuscript.

- L237: Remove "pressures"

Response: Thank you for your comment. We have changed in the revise manuscript.

Discussion

- L241: "with four other related Caprifoliaceae species" to "with four other closely related Lonicera species"

Response: Thank you for your comment. We have changed in the revise manuscript.

- L246: Remove "This means that the cp genomes of Lonicera species are conserved in length."

Response: Thank you for your comment. We have changed in the revise manuscript.

- L250: "sizes in" to "sizes of"

Response: Thank you for your comment. We have changed in the revise manuscript.

- L251: "four related Lonicera" to "four closely related Lonicera"

Response: Thank you for your comment. We have changed in the revise manuscript.

- L252: Remove "However, the gene types of the Lonicera cp genome were different." What are gene types? This sentence is too vague.

Response: Thank you for your comment. We have changed in the revise manuscript.

- L272: "preference is" to "preference was"

Response: Thank you for your comment. We have changed in the revise manuscript.

Reviewer: 2

The manuscript is well-organized and reports a study that uses contemporary methods to thoroughly examine the newly sequenced plastome of Lonicera ruprechtiana in the context of other complete plastomes from Caprifoliaceae family. This is a generally valuable study, like many other complete plastome studies that are currently published.

My main concern about the paper is on the assembly stats that are not adequately provided in the results, what was the N50 and the steps involved. Did The authors obtained a single contig in the with metaspades? I also hadn’t seen the use of the kraker for filtering cp and mt reads. The author really should clarify all of this issue. 

Response: Thank you for your important opinion. We have changed the assembly result part in the revise manuscript. The detail changes as follow:

The Illumina sequencing platform produced 3,059 Mb raw data. After identifies mitochondrial and chloroplast sequences in original data by Kraken2 software, 1,525,022 organelle reads were acquired. After filtered, 2,268 Mb clean reads with a Q20 value of 96.6% were obtained. The metaSPAdes software was used to assemble the Clean Illumina reads. According to the software assembly results, there were 548 non-redundant contigs with 1,060,153 bp in length. The N50 value was 6,924 bp. Further analysis of the assembly results based on the reference genome using BLASTn software, we got a single contig. Then, we used the BLAST+ (2.9.0) software to analyze the covariance between L. ruprechtiana and the corresponding reference genome (S1 Fig). The result showed L. ruprechtiana and L. sachalinensis have strong covariance relationship (S1 Fig). Finally, we successfully construct the complete cp genomes of L. ruprechtiana.

I also think that PCR should have been done for verifying the junctions followed by Sanger sequencing. This would have helped in justifying the assembly.

If PCR was not done, I recommend the authors the needs to include a coverage analysis of the assembly to ensure accurate assembly of the chloroplast genome. Please include this important analysis in the paper. 

Response: Thank you for your important opinion. we have done a coverage analysis of the assembly to ensure accurate assembly of the chloroplast genome. We used the blast+(2.9.0) software to analyze the covariance between L. ruprechtiana and the corresponding reference genome. The graph of covariance results is shown as Fig S1.

S1_Fig. Covariance plots. Red is the result of the covariance in the same direction; blue is the result of the covariance in the opposite direction.

In addition, since non-coding region are also powerful for phylogenetic resolution, were these regions tested in this case? If yes, was the result consistent with the one presented in the publication?

Response: Thank you for your important opinion. We did not use non-coding regions for the evolutionary tree, and as you suggested we tried to select non-coding sequences for the evolutionary tree, but since half of the genome is non-coding sequences, it was difficult to obtain sequences that occur in all 23 species. 

The authors should also clarify the number of genes used to construct the phylogenetic tree. 

Response: Thank you for your comment. We have changed in the revise manuscript.

The detail changes as follow:

Only the homologous CDs (coding gene sequences) were used to construct phylogenetic tree to reduce data redundancy. A total of 68 homologous CDs, including psbB, psbA, ndhB, rps7, rps12, psbA, matK, rps16, psbK, psbI, atpA, atpF, atpH, atpI, rps2, rpoC2, rpoC1, rpoB, petN, psbM, psbD, psbC, psbZ, rps14, ycf3, rps4, ndhJ, ndhC, atpE, atpB, rbcL, psaL, ycf4, cemA, petA, psbJ, psbF, psbE, petL, petG, psaJ, rpl33, rpl20, psbB, psbT, psbN, psbH, petB, petD, rps11, rpl36, infA, rps8, rpl14, rpl16, rpl22, rps19, rpl2, ndhF, rpl32, ccsA, ndhD, psaC, ndhE, ndhG, ndhI, ndhA, and ndhH, were used to determine the phylogenetic relationship.

The figures are also low quality and really should be improved.

Response: Thank you for your comment. 

We have changed our figures to meet PLOS ONE's style requirements.

---

## [Decision Letter · Decision Letter 1]

14 Dec 2021

PONE-D-21-23416R1Characterization of the chloroplast genome of Lonicera ruprechtiana Regel and comparison with other selected species of CaprifoliaceaePLOS ONE

Dear Dr. Weng,

Thank you for submitting your manuscript to PLOS ONE. After careful consideration, we feel that it has merit but does not fully meet PLOS ONE’s publication criteria as it currently stands. Therefore, we invite you to submit a revised version of the manuscript that addresses the points raised during the review process.

Additional authors' intervention regarding the phylogenetic analysis is required according to the Reviewer's recommendations. Moreover, the associated text should be considerably improved as it contains numerous inconsistencies. Generally, the whole manuscript should be thoroughly revised regarding proper language usage.

We look forward to receiving your revised manuscript.

Kind regards,

Branislav T. Šiler, Ph.D.

Academic Editor

PLOS ONE

Journal Requirements:

Reviewers' comments:

Reviewer's Responses to Questions

**Comments to the Author**

1. If the authors have adequately addressed your comments raised in a previous round of review and you feel that this manuscript is now acceptable for publication, you may indicate that here to bypass the “Comments to the Author” section, enter your conflict of interest statement in the “Confidential to Editor” section, and submit your "Accept" recommendation.

Reviewer #1: (No Response)

Reviewer #2: All comments have been addressed

2. Is the manuscript technically sound, and do the data support the conclusions?

Reviewer #1: Partly

Reviewer #2: Yes

3. Has the statistical analysis been performed appropriately and rigorously? 

Reviewer #1: No

Reviewer #2: Yes

4. Have the authors made all data underlying the findings in their manuscript fully available?

Reviewer #1: Yes

Reviewer #2: Yes

5. Is the manuscript presented in an intelligible fashion and written in standard English?

Reviewer #1: No

Reviewer #2: Yes

6. Review Comments to the Author

Reviewer #1: Characterization of the chloroplast genome of Lonicera ruprechtiana Regel and comparison with other selected species of Caprifoliaceae

This paper presents the chloroplast genome of Lonicera ruprechtiana and conducts a few comparative genomic analyses (i.e., structure, selection, pairwise alignment) between L. ruprechtiana and other members of Lonicera and Caprifoliaceae, more broadly. The authors provide a valuable genomic resource for future studies of the Caprifoliaceae, particularly repetitive elements that may be used for population genetics. The authors have addressed most of the comments raised by reviewers, but the language is not particularly strong. I therefore have only minor comments about the analyses but provide numerous suggestions to improve the readability of the manuscript.

----------------

General comments

----------------

1. Phylogenetic analysis. There is still a major issue with the rooting of the phylogeny presented and the way the phylogeny is discussed. Weigela should be the root, as in all other phylogenetic analyses of Caprifoliaceae prior to this study. The members of the Caprifolieae (Lonicera, Triosteum, and Heptacodium) should form a clade sister to the a clade containing Dipelta and Patrinia, with Weigela as the outgroup. The authors should comment on whether or not the phylogeny they present is consistent with previous work. Do the clades recovered make sense in comparison to the numerous plastid phylogenies for the Dipsacales have been published over the past 20 years?

The language surrounding phylogenetics is still imprecise. Phylogenies are not generally used to classify clusters of species (although this does occur in pop gen and microbial studies), they represent hypotheses of the relatedness of the samples they contain. In lines 240-242, the author's incorrectly state that L. ruprechtiana is sister to 3 taxa. In fig 3 it is clear that L. ruprechtiana is sister to L. insularis. This needs to be revised. I would caution the authors from placing too much stock into this phylogeny because the sampling is very poor and this is almost certainly not a true sister relationship.

I would also suggest moving the long list of genes from lines 132-137 into a table that shows missing data (i.e., which species you were able to gather which sequences from). In one of the comments to the reviewers the authors state that not all loci were recovered for all taxa, but we do not have any estimates of missing data.

2. Selection analysis. More details need to be filled in the Methods and statistics need to be included in the results. What model was used? Did you exclude any codon positions? What window size did you use? Why did you choose the species that you did? They are very closely related and therefore have fewer mutations to consider. A comparison between species with more varying relationships might be more informative. Also, there are no test statistics included here. Because it is very difficult to recover 1 exactly (Ka = Ks), you will always classify genes as under one form or another of selection, when, in reality, the number is not distinguishable from 1. I would like to see test statistics associated with these estimates. The authors state that differences were found when comparing to different species for some genes, due to "evolutionary differences"; what are these difference? Are there reasons we might think these species/genes have ben subject to different selection regimes?

3. Pairwise alignment (Fig. S1). The results shown in Fig S1 should be refered to a pairwise alignment, not covariance, in the text. If I understand the pairwise alignment (synteny) analysis in Fig S1, the genomes mostly preserve synteny, however there is one large inverted repeat that should be noted in the results. Typically in pairwise aligment plots like this, the red line should be broken where the blue lines are. What this plot seems to suggest is an inverted repeat. Is this correct? This should be discussed more in the results. See Fig. 1 of this paper on mammalian microinversions https://www.pnas.org/content/103/52/19824/tab-figures-data

-------------------

Line edits/comments

-------------------

Abstract

- L17-18: "The chloroplast (cp) genome is a powerful tool for resolving genome evolution." This statement is tautological and should be removed.

- L27: "IR" needs to be defined before the abbreviation is used

- There is a mix throughout the abstract of the present and past tense. This should be resolved so that the results are written in the past tense.

Introduction

- L39: "The Lonicera genus" to "Lonicera"

- L68-69: "Cp sequences have become a useful and powerful tool for revealing plant phylogenies [16]." is redundant with the two preceding sentences and should be removed.

- L70: Remove "phylogenomic"

- L77: "for the future genetic study" to "for future genetic studies"

Materials and Methods

- Throughout the M&M you do not need to write "software" after the name of the software

- L110: "of the related Lonicera" to "of Lonicera"

- L127-128: "between L. ruprechtiana and the other 22 cp genome sequences of the Caprifoliaceae" to "using the de novo L. ruprechtiana cp genome and 22 cp genomes from across the Caprifoliaceae"

- L137: "determine the phylogenetic relationship" to "determine phylogenetic relationships"

- L143: delete "in the Lonicera genus"

- L147: "divergence among the entire cp genome among these related species" to "divergence across entire cp genomes"

- I think it would be better to remove the species names from the subtitle "Synonymous and non-synonymous substitution rate calculations of L. ruprechtiana, L. ferdinandi, L. vesicaria, L. maackii, and L. insularis" and instead state the pairwise comparisons done in the body of the paragraph.

- L153: "close" to "closely related"

Results

- It is not important to repeatedly mention that you used an Illumina platform, and I would suggest removing the repeated mentions of it

- As in the M&M, you should not write "software" after the name of each program

- L158: "identifies" to "identifying"

- L159: "in original data by" to "using"

- L160: "organelle" to "organellar" and "filtered" to "filtering"

- L162-162: "contigs with 1,060,153 bp in length. The N50 value was 6,924 bp." to "contigs totaling 1,060,153 bp in length and with an N50 of 6,924 bp."

- L164: What is the reference genome you mention here?

- L164-167: This section on comparative genomics should be moved to the subsection below on phylogenetics and comparative genomics. You should also state which program you used to the the pairwise alignment between the two genomes; BLAST+ is the suite of all BLAST-related software, not an actual program iteself. Also see notes above on Fig. S1.

- L167-168: Delete "Finally, we successfully construct the complete cp genomes of L. ruprechtiana."

- L169: Are the IRA and IRB the repeats shown in Fig S1? If so, it should be referenced. You should also be referencing Fig. 1 when refering to these loci because it shows their positions and orientations.

- L177: "double" to "two"

- L190 (Table 1 caption): Delete "The detail"

- Table 1: "Construction of cp genome" to "Lengths of major regions"

- L208: "ranged" to "ranging"

- L209: "ranged" to "ranging"

- L213: "contributed by" to "contained in"

- L214: "intergenic region or partly in the gene spacer region" to "intergenic or spacer regions"

- L216 (Table 4 caption): Delete "Detail information of". I don't think Table 4 is important enough to take up so much room in the main text. I would suggest making it supplemental.

- L219: "sequences" to "sequences of the" and "produced" to "contained"

- L220-221: "Among these, the codon of leucine had the highest usage frequency, at 10.7%, while the usage frequency of cysteine was only 1.1% (S2 Table; Fig 2A)." to "Among these, leucine had the highest usage frequency, at 10.7%, while cysteine was least freuqent, at only 1.1% (S2 Table; Fig 2A)."

- L223: "with" to "had"

- L224: "in the L. ruprechtiana." to "by L. ruprechtiana"

- L225: "were" to "was"

- L227-230: The figure caption should be used to explain the figure, not tell results. I suggest simplifying the capture to "Percentage of amino acids of the L. ruprechtiana chloroplast (cp) genome (A) and the ending patterns of biased-usage codons (RSUC>1) (B)."

- L234-237: Rephrase "To further understand the phylogenetic relationships of L. ruprechtiana, 68 homologous protein-coding genes of 22 cp genome sequences of the Caprifoliaceae were downloaded from NCBI database to build the phylogenetic tree. The phylogenetic tree was created using MEGA7 [25] with 1,000 bootstrap replicates." to "To further understand the phylogenetic placement of L. ruprechtiana, 68 homologous protein-coding genes of 22 Caprifoliaceae cp genome sequences downloaded from NCBI were used to estimate a phylogeny using MEGA7 [25] with 1,000 bootstrap replicates (Fig. 3)."

- Delete lines 237-240 "To show... 2 distinct clusters." The statement that the authors only show a cladogram should be left for the figure caption, and see my notes on clusters above.

- L240: Delete "in the Lonicera cluster". Again, Lonicera a clade, not a cluster. The sentence also references Fig. 3 twice, and should only reference it once.

- L244-245: Delete "The maximum-likelihood tree shows the two distinct clusters."

- L246: Delete "in the figure"

- L250: "of five" to "of the five"

- L251: "the mVISTA program" to "mVISTA"

- L253: "mean" to "show" and "are" to "were"

- L254: "are consistent with those of" to "consistent with"

- L262: what does "were relatively more highly" mean? More highly what? It seems like something is missing here

- L275-276: "were downloaded from GenBank and analyzed" to "are shown"

- L277: "negative-strand" to "reverse-strand"

- L283-284: "the evolution of genome and selection genes under pressure" to "selection pressure on genes and genomes"

- L294: "genes are strong" to "genes are under strong"

- L297: "the Illumina" to "an Illumina". There are many Illumina platforms

- L298: delete "others"

- L298-303 are meandering and repetitive. Possibly rephrase like "The complete cp genome of L. ruprechtiana showed a typical quadripartite cycle of 154,611 bp in length length, comparable to that of published Lonicera species cp genomes (154,513–155,346) (Fig. 1, Tables 1 and S1) [17]."

- L303: "variable" to "variation"

- 322: "the sequence of ycf2 gene have the highest variable" to "ycf2 is one of the most variable genes"

- L338: "the Illumina" to "an Illumina"

- L341: delete "information"

- L344: "had a close relationship with" to "is closely related to". See my note above; I caution over interpretation of this phylogeny

Supplement

- L469: "List of the cp genome of 23 species used for phylogenetic analysis." to "List of the 23 cp genomes used for phylogenetic analysis."

- L472: "rate" to "rates"

Reviewer #2: The authors have addressed all my concerns and therefore I support publication without further

changes.

7. PLOS authors have the option to publish the peer review history of their article (what does this mean?). If published, this will include your full peer review and any attached files.

Reviewer #1: No

Reviewer #2: No

---

## [Author Response · Author response to Decision Letter 1]

3 Jan 2022

December 31, 2021

Dr Branislav T. Šiler

Academic Editor 

PLOS ONE 

Dear Editor and Reviewers,

RE: Response to Decision Letter for Manuscript ID. PONE-D-21-23416R1

First, thank you for your comments. We have now revised our manuscript further in accordance with your recommendations. 

The following is the response to the comments of the academic editor and the reviewers with a detailed and clear point-by-point response.

We are looking forward to your positive decision. 

Sincerely yours,

Qingbei Weng, 

Ph.D, Professor

Department of Biotechnology

School of Life Sciences, 

Guizhou Normal University, Guiyang, China

Manuscript ID. PONE-D-21-23416R1

Dear Dr. Weng,

Thank you for submitting your manuscript to PLOS ONE. After careful consideration, we feel that it has merit but does not fully meet PLOS ONE’s publication criteria as it currently stands. Therefore, we invite you to submit a revised version of the manuscript that addresses the points raised during the review process.

Additional authors' intervention regarding the phylogenetic analysis is required according to the Reviewer's recommendations. Moreover, the associated text should be considerably improved as it contains numerous inconsistencies. Generally, the whole manuscript should be thoroughly revised regarding proper language usage.

We look forward to receiving your revised manuscript.

Kind regards,

Branislav T. Šiler, Ph.D.

Academic Editor

PLOS ONE

Journal Requirements:

Response: Thank you for your important comment. We have reviewed our reference list to ensure that it is complete and correct. We are not cited papers that have been retracted. There is one Chinese paper. Reviewer 1 had said there are multiple claims about the phytochemistry of L. ruprechtiana in comparison to L. japonica in this paragraph but none of them are cited. What is the evidence for these claims?” So, we add a Chinese paper:

7: Zhu D, Zhao X, Dai L, Ji S. Experimental study on the pharmacology of Lonicera ruprechtiana Regel. Theory and Practice of Chinese Medicine. 2003; 2003: 111-112. (in chinese) 

This article was published in Chinese magazine, so I cited with “in Chinese” at the end of this paper.

We changed another chinese paper: 

7: Wang G, Zhou X, Cui J, Zhao X, Yang X. Iridoid compounds in the buds of Lonicera ruprechtiana Regel. Chinese Journal of Medicinal Chemistry. 2009; 19: 206-208. (in chinese) (Web address: https://kns.cnki.net/kns8/defaultresult/index)

Reviewer(s)' Comments to Author:

Reviewer #1: Characterization of the chloroplast genome of Lonicera ruprechtiana Regel and comparison with other selected species of Caprifoliaceae

This paper presents the chloroplast genome of Lonicera ruprechtiana and conducts a few comparative genomic analyses (i.e., structure, selection, pairwise alignment) between L. ruprechtiana and other members of Lonicera and Caprifoliaceae, more broadly. The authors provide a valuable genomic resource for future studies of the Caprifoliaceae, particularly repetitive elements that may be used for population genetics. The authors have addressed most of the comments raised by reviewers, but the language is not particularly strong. I therefore have only minor comments about the analyses but provide numerous suggestions to improve the readability of the manuscript.

----------------

General comments

----------------

1. Phylogenetic analysis. There is still a major issue with the rooting of the phylogeny presented and the way the phylogeny is discussed. Weigela should be the root, as in all other phylogenetic analyses of Caprifoliaceae prior to this study. The members of the Caprifolieae (Lonicera, Triosteum, and Heptacodium) should form a clade sister to a clade containing Dipelta and Patrinia, with Weigela as the outgroup. The authors should comment on whether or not the phylogeny they present is consistent with previous work. Do the clades recovered make sense in comparison to the numerous plastid phylogenies for the Dipsacales have been published over the past 20 years?

Response: Thank you for your important comment. We have changed phylogenetic analyses using Weigela as root, and also changed relative result parts in the revise manuscript. The detail changes as follow:

“We used Weigela as the root, as in other phylogenetic analyses of Caprifoliaceae prior to this study[17, 18]. To show the relationships between L. ruprechtiana and other 22 family members, the phylogenetic tree is a cladogram and branch lengths we not infered (Fig 3). The members of the Caprifolieae (Lonicera, Triosteum, and Heptacodium) form a clade sister to a clade containing Dipelta and Patrinia, with Weigela as the outgroup (Fig 3). As shown in Fig. 3, L. ruprechtiana was sister to L. insularis with a bootstrap support of 100%.”

17. Wang HX, Liu H, Moore MJ, Landrein S, Liu B, Zhu ZX, Wang HF. Plastid phylogenomic insights into the evolution of the Caprifoliaceae s.l. (Dipsacales). Mol Phylogenet Evol. 2020; 142: 106641.

18. Lee AK, Gilman IS, Srivastav M, Lerner AD, Donoghue MJ, Clement WL. Reconstructing Dipsacales phylogeny using Angiosperms353: issues and insights. Am J Bot. 2021; 108: 1122-1142.

The language surrounding phylogenetics is still imprecise. Phylogenies are not generally used to classify clusters of species (although this does occur in pop gen and microbial studies), they represent hypotheses of the relatedness of the samples they contain. In lines 240-242, the author's incorrectly state that L. ruprechtiana is sister to 3 taxa. In fig 3 it is clear that L. ruprechtiana is sister to L. insularis. This needs to be revised. I would caution the authors from placing too much stock into this phylogeny because the sampling is very poor and this is almost certainly not a true sister relationship.

Response: Thank you for your comment. We have changed in the revise manuscript. The detail changes as follow:

“We used Weigela as the root, as in all other phylogenetic analyses of Caprifoliaceae prior to this study[17, 18]. To show the relationships between L. ruprechtiana and other 22 family members, the phylogenetic tree is a cladogram and branch lengths we not infered (Fig 3). The members of the Caprifolieae (Lonicera, Triosteum, and Heptacodium) form a clade sister to a clade containing Dipelta and Patrinia, with Weigela as the outgroup (Fig 3). As shown in Fig. 3, L. ruprechtiana was sister to L. insularis with a bootstrap support of 100%.”

I would also suggest moving the long list of genes from lines 132-137 into a table that shows missing data (i.e., which species you were able to gather which sequences from). In one of the comments to the reviewers the authors state that not all loci were recovered for all taxa, but we do not have any estimates of missing data.

Response: Thank you for your comment. We have added Table S2 to show which species were able to gather which sequences from. 

“A total of 68 homologous CDs (S2 Table) were used to determine phylogenetic relationships.”

2. Selection analysis. More details need to be filled in the Methods and statistics need to be included in the results. What model was used? Did you exclude any codon positions? What window size did you use? Why did you choose the species that you did? They are very closely related and therefore have fewer mutations to consider. A comparison between species with more varying relationships might be more informative. Also, there are no test statistics included here. Because it is very difficult to recover 1 exactly (Ka = Ks), you will always classify genes as under one form or another of selection, when, in reality, the number is not distinguishable from 1. I would like to see test statistics associated with these estimates. The authors state that differences were found when comparing to different species for some genes, due to "evolutionary differences"; what are this difference? Are there reasons we might think these species/genes have been subject to different selection regimes?

More details need to be filled in the Methods and statistics need to be included in the results. What model was used? Did you exclude any codon positions? What window size did you use? Why did you choose the species that you did?

Response: Thank you for your comment. We have changed in the revise manuscript. The detail changes as follow:

Filled related parameters in the Methods:

“In addition, the mVISTA program (http://genome.lbl.gov/vista/mvista/submit.shtml) was used to compare to divergence across entire cp genomes with default settings (window size, 100bp; RepeatMasker, do not mask; RankVISTA probability threshold, 0.5).”

They are very closely related and therefore have fewer mutations to consider. A comparison between species with more varying relationships might be more informative.

Response: Thank you for your useful comment “A comparison between species with more varying relationships might be more informative”, In this work, we investigate divergence in the cp genome between L. ruprechtiana and the four other closely related species (L. ferdinandi, L. vesicaria, L. maackii, and L. insularis) because of these species had closely relationship. We should do the comparison between species with more varying relationships.

Also, there are no test statistics included here. Because it is very difficult to recover 1 exactly (Ka = Ks), you will always classify genes as under one form or another of selection, when, in reality, the number is not distinguishable from 1. I would like to see test statistics associated with these estimates.

Response: Thank you for your comment. We have added the test statistics in Table S5 (Synonymous (Ks) and non-synonymous (Ka) substitution rates of common coding genes between L. ruprechtiana and other four closed related species).

The authors state that differences were found when comparing to different species for some genes, due to "evolutionary differences"; what are this difference? Are there reasons we might think these species/genes have been subject to different selection regimes?

Response: Thank you for your comment. We have changed in the revise manuscript. The detail changes as follow:

“Some genes (atpE, atpF, matK, ndhB, petB, petD, psaI, and rpl14) had different Ka/Ks values (>1 or <1) under different comparisons (S5 Table), possibly due to these species/genes have been subject to different selection regimes.”

3. Pairwise alignment (Fig. S1). The results shown in Fig S1 should be refered to a pairwise alignment, not covariance, in the text. If I understand the pairwise alignment (synteny) analysis in Fig S1, the genomes mostly preserve synteny, however there is one large inverted repeat that should be noted in the results. Typically, in pairwise aligment plots like this, the red line should be broken where the blue lines are. What this plot seems to suggest is an inverted repeat. Is this correct? This should be discussed more in the results. See Fig. 1 of this paper on mammalian microinversions https://www.pnas.org/content/103/52/19824/tab-figures-data

Response: Thank you for your comment. We have changed in the revise manuscript. The detail changes as follow:

“Then, we used the BLAST+ (Version, 2.9.0) software to pairwise alignment between L. ruprechtiana and the corresponding reference genome (S1 Fig). As shown in S1 Fig, the genomes mostly preserve synteny, however there was one large inverted repeat (IRA and IRB regions, Fig 1); consistent with earlier research about mammalian evolution [28]”

28: Chaisson MJ, Raphael BJ, Pevzner PA. Microinversions in mammalian evolution. Proc Natl Acad Sci U S A. 2006; 103: 19824-9.

We also changed the figure legend of figure S1: “S1 Fig. Pairwise alignment plots. Red is the result in the same direction; blue is the result in the opposite direction”.

-------------------

Line edits/comments

-------------------

Abstract

- L17-18: "The chloroplast (cp) genome is a powerful tool for resolving genome evolution." This statement is tautological and should be removed.

Response: Thank you for your comment. We have changed in the revise manuscript.

- L27: "IR" needs to be defined before the abbreviation is used

Response: Thank you for your comment. We have changed in the revise manuscript.

- There is a mix throughout the abstract of the present and past tense. This should be resolved so that the results are written in the past tense.

Response: Thank you for your comment. We have changed in the revise manuscript.

The results in abstract are written in the past tense.

Introduction

- L39: "The Lonicera genus" to "Lonicera"

Response: Thank you for your comment. We have changed in the revise manuscript.

- L68-69: "Cp sequences have become a useful and powerful tool for revealing plant phylogenies [16]." is redundant with the two preceding sentences and should be removed.

Response: Thank you for your comment. We have changed in the revise manuscript.

We have removed this sentence.

- L70: Remove "phylogenomic"

Response: Thank you for your comment. We have changed in the revise manuscript.

- L77: "for the future genetic study" to "for future genetic studies"

Response: Thank you for your comment. We have changed in the revise manuscript.

Materials and Methods

- Throughout the M&M you do not need to write "software" after the name of the software

Response: Thank you for your comment. We have changed in the revise manuscript.

- L110: "of the related Lonicera" to "of Lonicera"

Response: Thank you for your comment. We have changed in the revise manuscript.

- L127-128: "between L. ruprechtiana and the other 22 cp genome sequences of the Caprifoliaceae" to "using the de novo L. ruprechtiana cp genome and 22 cp genomes from across the Caprifoliaceae"

Response: Thank you for your comment. We have changed in the revise manuscript.

- L137: "determine the phylogenetic relationship" to "determine phylogenetic relationships"

Response: Thank you for your comment. We have changed in the revise manuscript.

- L143: delete "in the Lonicera genus"

Response: Thank you for your comment. We have changed in the revise manuscript.

- L147: "divergence among the entire cp genome among these related species" to "divergence across entire cp genomes"

Response: Thank you for your comment. We have changed in the revise manuscript.

- I think it would be better to remove the species names from the subtitle "Synonymous and non-synonymous substitution rate calculations of L. ruprechtiana, L. ferdinandi, L. vesicaria, L. maackii, and L. insularis" and instead state the pairwise comparisons done in the body of the paragraph.

Response: Thank you for your comment. We have changed in the revise manuscript.

We have removed the species names.

- L153: "close" to "closely related"

Response: Thank you for your comment. We have changed in the revise manuscript.

Results

- It is not important to repeatedly mention that you used an Illumina platform, and I would suggest removing the repeated mentions of it

- As in the M&M, you should not write "software" after the name of each program

Response: Thank you for your comment. We have changed in the revise manuscript.

- L158: "identifies" to "identifying"

Response: Thank you for your comment. We have changed in the revise manuscript.

- L159: "in original data by" to "using"

Response: Thank you for your comment. We have changed in the revise manuscript.

- L160: "organelle" to "organellar" and "filtered" to "filtering"

Response: Thank you for your comment. We have changed in the revise manuscript.

- L162-162: "contigs with 1,060,153 bp in length. The N50 value was 6,924 bp." to "contigs totaling 1,060,153 bp in length and with an N50 of 6,924 bp."

Response: Thank you for your comment. We have changed in the revise manuscript.

- L164: What is the reference genome you mention here?

Response: Thank you for your comment. It means Lonicera sachalinensis (GenBank accession: MH028742), the reference sequences used for assembly the cp genome of L. ruprechtiana. We have changed in the revise manuscript. The detail changes as follow:“Further analysis of the assembly results based on the reference genome (Lonicera sachalinensis, the reference sequences used for assembly the cp genome of L. ruprechtiana) using BLASTn, we got a single contig.”

- L164-167: This section on comparative genomics should be moved to the subsection below on phylogenetics and comparative genomics. You should also state which program you used to the pairwise alignment between the two genomes; BLAST+ is the suite of all BLAST-related software, not an actual program iteself. Also see notes above on Fig. S1.

Response: Thank you for your comment. This section on comparative genomics state that how to assembly the cp genome of L. ruprechtiana, it may be more suitable in the result part of Cp genome assembly and genome features.

We have added the version number of Blast+ in the revise manuscript.

- L167-168: Delete "Finally, we successfully construct the complete cp genomes of L. ruprechtiana."

Response: Thank you for your comment. We have changed in the revise manuscript.

- L169: Are the IRA and IRB the repeats shown in Fig S1? If so, it should be referenced. You should also be referencing Fig. 1 when refering to these loci because it shows their positions and orientations.

Response: Thank you for your comment. IRA and IRB are the repeats shown in Fig S1. We have referenced in the revise manuscript.

- L177: "double" to "two"

Response: Thank you for your comment. We have changed in the revise manuscript.

- L190 (Table 1 caption): Delete "The detail"

Response: Thank you for your comment. We have changed in the revise manuscript.

- Table 1: "Construction of cp genome" to "Lengths of major regions"

Response: Thank you for your comment. We have changed in the revise manuscript.

- L208: "ranged" to "ranging"

Response: Thank you for your comment. We have changed in the revise manuscript.

- L209: "ranged" to "ranging"

Response: Thank you for your comment. We have changed in the revise manuscript.

- L213: "contributed by" to "contained in"

Response: Thank you for your comment. We have changed in the revise manuscript.

- L214: "intergenic region or partly in the gene spacer region" to "intergenic or spacer regions"

Response: Thank you for your comment. We have changed in the revise manuscript.

- L216 (Table 4 caption): Delete "Detail information of". I don't think Table 4 is important enough to take up so much room in the main text. I would suggest making it supplemental.

Response: Thank you for your comment. We have changed in the revise manuscript.

We have making Table 4 supplemental (S3 Table). 

- L219: "sequences" to "sequences of the" and "produced" to "contained"

Response: Thank you for your comment. We have changed in the revise manuscript.

- L220-221: "Among these, the codon of leucine had the highest usage frequency, at 10.7%, while the usage frequency of cysteine was only 1.1% (S2 Table; Fig 2A)." to "Among these, leucine had the highest usage frequency, at 10.7%, while cysteine was least freuqent, at only 1.1% (S2 Table; Fig 2A)."

Response: Thank you for your comment. We have changed in the revise manuscript.

- L223: "with" to "had"

Response: Thank you for your comment. We have changed in the revise manuscript.

- L224: "in the L. ruprechtiana." to "by L. ruprechtiana"

Response: Thank you for your comment. We have changed in the revise manuscript.

- L225: "were" to "was"

Response: Thank you for your comment. We have changed in the revise manuscript.

- L227-230: The figure caption should be used to explain the figure, not tell results. I suggest simplifying the capture to "Percentage of amino acids of the L. ruprechtiana chloroplast (cp) genome (A) and the ending patterns of biased-usage codons (RSUC>1) (B)."

Response: Thank you for your comment. We have changed in the revise manuscript.

- L234-237: Rephrase "To further understand the phylogenetic relationships of L. ruprechtiana, 68 homologous protein-coding genes of 22 cp genome sequences of the Caprifoliaceae were downloaded from NCBI database to build the phylogenetic tree. The phylogenetic tree was created using MEGA7 [25] with 1,000 bootstrap replicates." to "To further understand the phylogenetic placement of L. ruprechtiana, 68 homologous protein-coding genes of 22 Caprifoliaceae cp genome sequences downloaded from NCBI were used to estimate a phylogeny using MEGA7 [25] with 1,000 bootstrap replicates (Fig. 3)."

Response: Thank you for your comment. We have changed in the revise manuscript.

- Delete lines 237-240 "To show... 2 distinct clusters." The statement that the authors only show a cladogram should be left for the figure caption, and see my notes on clusters above.

Response: Thank you for your comment. We have changed in the revise manuscript.

- L240: Delete "in the Lonicera cluster". Again, Lonicera a clade, not a cluster. The sentence also references Fig. 3 twice, and should only reference it once.

Response: Thank you for your comment. We have changed in the revise manuscript.

- L244-245: Delete "The maximum-likelihood tree shows the two distinct clusters."

Response: Thank you for your comment. We have changed in the revise manuscript.

- L246: Delete "in the figure"

Response: Thank you for your comment. We have changed in the revise manuscript.

- L250: "of five" to "of the five"

Response: Thank you for your comment. We have changed in the revise manuscript.

- L251: "the mVISTA program" to "mVISTA"

Response: Thank you for your comment. We have changed in the revise manuscript.

- L253: "mean" to "show" and "are" to "were"

Response: Thank you for your comment. We have changed in the revise manuscript.

- L254: "are consistent with those of" to "consistent with"

Response: Thank you for your comment. We have changed in the revise manuscript.

- L262: what does "were relatively more highly" mean? More highly what? It seems like something is missing here

Response: Thank you for your comment. We have changed in the revise manuscript.

The detail changes as follow: “were relatively more conservative than”

- L275-276: "were downloaded from GenBank and analyzed" to "are shown"

Response: Thank you for your comment. We have changed in the revise manuscript.

- L277: "negative-strand" to "reverse-strand"

Response: Thank you for your comment. We have changed in the revise manuscript.

- L283-284: "the evolution of genome and selection genes under pressure" to "selection pressure on genes and genomes"

Response: Thank you for your comment. We have changed in the revise manuscript.

- L294: "genes are strong" to "genes are under strong"

Response: Thank you for your comment. We have changed in the revise manuscript.

- L297: "the Illumina" to "an Illumina". There are many Illumina platforms

Response: Thank you for your comment. We have changed in the revise manuscript.

- L298: delete "others"

Response: Thank you for your comment. We have changed in the revise manuscript.

- L298-303 are meandering and repetitive. Possibly rephrase like "The complete cp genome of L. ruprechtiana showed a typical quadripartite cycle of 154,611 bp in length length, comparable to that of published Lonicera species cp genomes (154,513–155,346) (Fig. 1, Tables 1 and S1) [17]."

Response: Thank you for your comment. We have changed in the revise manuscript.

- L303: "variable" to "variation"

Response: Thank you for your comment. We have changed in the revise manuscript.

- 322: "the sequence of ycf2 gene have the highest variable" to "ycf2 is one of the most variable genes"

Response: Thank you for your comment. We have changed in the revise manuscript.

- L338: "the Illumina" to "an Illumina"

Response: Thank you for your comment. We have changed in the revise manuscript.

- L341: delete "information"

Response: Thank you for your comment. We have changed in the revise manuscript.

- L344: "had a close relationship with" to "is closely related to". See my note above; I caution over interpretation of this phylogeny

Response: Thank you for your comment. We have changed in the revise manuscript.

Supplement

- L469: "List of the cp genome of 23 species used for phylogenetic analysis." to "List of the 23 cp genomes used for phylogenetic analysis."

Response: Thank you for your comment. We have changed in the revise manuscript.

- L472: "rate" to "rates"

Response: Thank you for your comment. We have changed in the revise manuscript.

Reviewer #2: The authors have addressed all my concerns and therefore I support publication without further

changes.

Response: Thank you for your positive comment.

---

## [Editor Report · Decision Letter 2]

6 Jan 2022

Characterization of the chloroplast genome of Lonicera ruprechtiana Regel and comparison with other selected species of Caprifoliaceae

PONE-D-21-23416R2

Dear Dr. Weng,

We’re pleased to inform you that your manuscript has been judged scientifically suitable for publication and will be formally accepted for publication once it meets all outstanding technical requirements.

Kind regards,

Branislav T. Šiler, Ph.D.

Academic Editor

PLOS ONE
---

## [Editor Report · Acceptance letter]

14 Jan 2022

PONE-D-21-23416R2 

Characterization of the chloroplast genome of *Lonicera ruprechtiana* Regel and comparison with other selected species of Caprifoliaceae 

Dear Dr. Weng:

I'm pleased to inform you that your manuscript has been deemed suitable for publication in PLOS ONE. Congratulations! Your manuscript is now with our production department. 

Kind regards, 

on behalf of

Dr. Branislav T. Šiler 

Academic Editor

PLOS ONE